# Investigating latent syphilis in HIV treatment-experienced Ethiopians and response to therapy

**Selamawit Girma[1]☯, Wondwossen Amogne[2]☯***

1 College of Health Sciences, School of Medicine, Department of Dermatovenerology, Addis Ababa University, Addis Ababa, Ethiopia, 2 College of Health Sciences, School of Medicine, Department of Internal Medicine, Addis Ababa University, Addis Ababa, Ethiopia

☯ These authors contributed equally to this work.
* wondwossen.amogne@aau.edu.et, wonamogne@yahoo.com

## Abstract

### Objectives

We investigated people with HIV (PWH) receiving combination antiretroviral therapy (cART) for latent syphilis infection prevalence, risk factors, treatment response, and neurosyphilis.

### Methods

A prospective follow-up study was conducted on PWH and latent syphilis. The cases were randomly assigned to receive either benzathine penicillin G (BPG) or doxycycline (DOXY), and the posttreatment response was evaluated after 12 and 24 months. The traditional algorithm was used for serodiagnosis, and a semi-quantitative rapid plasma reagin (RPR) test monitored disease activity and treatment effectiveness.

### Results

Of the 823 participants, 64.8% were women, and the mean age was 41.7±10 years. Thirty-one (3.8%) of the participants (22 males and nine females) had latent syphilis. The risk factors were male sex (aOR = 3.14), increasing age (aOR = 1.04 per year), and cART duration (aOR = 1.01 per month). Baseline RPR titers were: ≤1:4 in 19 (61.3%), between 1:8 and 1:32 in 10 (32.2%), and >1:32 in 2 (6.4%). None of the seven cerebrospinal fluid analyses supported a neurosyphilis diagnosis. In the 12th month of treatment, 27 (87.1%) had adequate serological responses, three (9.7%) had serological nonresponse, and one (3.2%) had treatment failure. Syphilis treatment was repeated in the last four cases with the alternative drug. In terms of adequate serologic response, both therapies were comparable at the 12th month, p = 0.37. All cases responded to treatment in the 24th month.

### Conclusion

In PWH receiving cART, latent syphilis occurred more in men than women, suggesting an investigation of sexual practices and the impact of antenatal syphilis screening. Syphilis disease activity reduces in the latent stage. Therefore, the routine cerebrospinal fluid analysis

**Data Availability Statement:** We have uploaded the data in spreadsheet in other files.

**Funding:** The research was supported by a Medical Education Partnership Initiative (MEPI) grant for junior faculty members (D43TW010143) obtained

from the US National Institutes of Health, Fogarty International Center. The funder had no role in the study design, conduct, data collection, analysis, decision to publish, or preparation of manuscript. The grant received was to conduct the study and does not cover expenses related to the manuscript preparation and publication.

**Competing interests:** The authors have declared that no competing interests exist.

contributes little to the diagnosis of asymptomatic neurosyphilis and the treatment success of latent syphilis. DOXY is an alternative to BPG, and cART improves serologic response to latent syphilis treatment.

## Background

In 2016 the World Health Organization (WHO) estimated that the prevalence of syphilis was 19.9 million, with an incidence of 6.3 million cases between 15 and 49 years of age [1]. Over 90% of these cases occurred in low-income countries where HIV is also most prevalent [2, 3]. As of 2018, the global incidence rate of syphilis is 1.7 cases per 1000 women and 1.6 per 1000 men [2]. In 2014, the Ethiopian national prevalence was 1.2% for syphilis and 2% for HIV infection, and syphilis was two times more common in HIV-positive individuals [4]. Unfortunately, there is no national data to show the prevalence trend of syphilis in people with HIV (PWH) [5], which is a critical gap given the complex interactions which exist between syphilis and HIV coinfection [5–7].

Early syphilis increases the risk of HIV acquisition and transmission [8, 9]. In addition, it affects the immunological and virological control of HIV infection, whereas HIV may affect the natural course, diagnosis, and treatment effectiveness of syphilis [10–18]. Several studies suggest that using combination antiretroviral therapy (cART) has improved syphilis treatment outcomes [19, 20]. HIV transmission is highly unlikely in PWH with suppressed viral load and no active syphilitic lesions [21, 22]. Nevertheless, there is evidence of an increased incidence of syphilis in PWH receiving combination cART consequent to sexual behavior [7, 23].

In contrast to early syphilis, the interaction between latent syphilis and HIV coinfection treated with cART is more nebulous. Serology is the mainstay of latent syphilis diagnosis as there are no clinical manifestations [24]. Indeed, many PWH are diagnosed with latent syphilis without the prior symptomatic disease [25, 26]. The latency duration is difficult to assess, making latent syphilis challenging to categorize as early, late, or unknown [27]. The nontreponemal (NTT) antibody titers are generally higher in early syphilis infection and correlate with the disease activity [24]. Syphilis is less active during the latent stage; however, it does not necessarily mean the disease is quiescent [28].

Neurosyphilis, which may occur at any stage of infection, develops in about 5% of cases with untreated infection [29, 30]. The risk of developing it increases several folds for HIV coinfected individuals, making them a critical risk group for monitoring [31]. In HIV coinfected patients with neurosyphilis, the median NTT antibody titer is significantly higher than those without neurosyphilis [32]. Still, Africa has limited data about neurosyphilis in PWH receiving cART, which is an evidence gap we seek to address here [33]. Syphilis treatment is critical to prevent subsequent clinical manifestations in PWH, but the optimal choice of antimicrobial agents to treat syphilis during HIV coinfection is controversial [34]. The guidelines recommend benzathine penicillin G (BPG) for latent syphilis, irrespective of HIV status. Oral doxycycline (DOXY) is an alternate option [24, 34]. However, there is sparse data about serological response following treatment with DOXY in PWH receiving cART [35].

Studies evaluating the serologic response to syphilis treatment in HIV coinfected individuals show conflicting results [15, 36]. A retrospective study [15] compared 129 PWH (28% early and 67% late syphilis) and 168 patients without HIV (92% early and 8% late syphilis) and found a fourfold drop in the rapid plasma reagin (RPR) titer occurred less frequently in PWH. The researchers suggested that in the context of HIV coinfection, syphilis may be at higher risk

of serological failure. A randomized controlled trial [36] evaluated the treatment response of early syphilis in patients with and without HIV. The result showed an increased likelihood of serologic failure with HIV coinfection; however, clinical failure was uncommon in both groups. Of note, all did not receive cART in both studies.

Given the lack of data on latent syphilis infection among PWH in Ethiopia, the need to resolve syphilis infection among PWH to prevent future complications, and the conflicting opinions about optimal treatment approaches, our study aims to evaluate latent syphilis prevalence, assess risk factors, investigate the prevalence of neurosyphilis, and compare treatment responses between BPG and DOXY in a cohort of PWH receiving cART in Ethiopia.

## Materials and methods

### Study setting

We conducted the study in Addis Ababa, the capital city of Ethiopia, from December 2018 to April 2021. The study sites were Tikur Anbessa specialized referral hospital (TASH), Alert referral hospital (ARH), Teklehaimanot health center (THC), and Kazanchis health center (KHC). The total numbers of PWH during the study period at TASH, ARH, THC, and KHC were 3600, 6595, 1328, and 2051 respectively. The serological tests for syphilis were performed by Arsho Medical Laboratory, Addis Ababa, Ethiopia. The Ethiopian National Accreditation Office accredited the lab (Faculty Accreditation number: M0051) per ISO 15189:2012, medical laboratory requirements for quality & competence.

### Study design

The study had a prospective follow-up design. First, we selected every third person from the HIV clinic registry to investigate latent syphilis. Next, the diagnosed cases were assigned to receive therapy with BPG or DOXY using a simple random sampling technique. Last, we followed the treatment responses for 12 and 24 months.

### Ethical consideration

We obtained informed written consent from the study participants. The institutional review board of the college of health sciences at the Addis Ababa University, Armauer Hansen Research Institute, and the Addis Ababa regional health bureau approved the study protocol.

### Participant inclusion and exclusion criteria

The inclusion criteria were 18 years of age or older, PWH on cART, and informed consent. The exclusion criteria were pregnancy, clinically recognizable primary, secondary, or tertiary syphilis manifestations, and history of severe penicillin allergy.

### The diagnosis of latent syphilis

The traditional algorithm for syphilis diagnosis was adopted; the results were reported as reactive and nonreactive. The screening was performed with a rapid plasma reagin (RPR) test and confirmed with the *Treponema pallidum* hemagglutination assay (TPHA). For measuring the strength of antibody response, the RPR titers were semi-quantitatively determined with the RPR card test. The highest dilution showing visible clumps was reported. Treatment effectiveness was assessed similarly with the RPR titer after 12 and 24 months of therapy. Biologic false-positive RPR (BFP-RPR) was defined as reactive RPR and nonreactive TPHA, while reactive RPR and TPHA test results in the absence of clinical manifestations described latent syphilis. Our study's diagnosis of latent syphilis includes early latent, late latent, and unknown

duration. The laboratory performed the initial qualitative testing with Carbogen, an RPR card test that uses carbon antigen for syphilis testing from Tulip Diagnostics (P) LTD, India. A reactive test result was confirmed with SEROCHECK-Tp$^{TM}$, a modified TPHA qualitative test from Tulip Diagnostics (P) LTD, India. The laboratory tests were carried out according to the manufacturers' directions. Those who had weakly reactive results were retested. All patients diagnosed with latent syphilis had a careful examination of the skin, oral cavity, perianal area, genital organs, cardiovascular, and nervous system examinations. A direct ophthalmoscope and otoscopic evaluations were included.

## The diagnosis of neurosyphilis

The diagnosis of neurosyphilis was confirmed by a reactive cerebrospinal fluid (CSF) venereal disease research laboratory (VDRL) test result or a CSF white blood cells (WBC) count above ten cells/mm$^3$. The CSF protein level was excluded due to other reasons for elevation in PWH. The indications for CSF analysis were: a/ serum RPR titer $\geq$ 1:32 b/ peripheral blood CD4 count $\leq$ 350 cells/mm$^3$, and HIV RNA level above 2.3 log/mL (or 200 copies/mL) c/ serological nonresponse or treatment failure after latent syphilis therapy and d/ unexplained persistent neurologic, ocular or otic symptoms or signs.

## Latent syphilis treatment

The latent syphilis cases received weekly intramuscular injections of BPG 2.4 million units for three consecutive weeks or oral DOXY 100 mg twice daily for 28 days. Study participants who refused BPG injections were given DOXY and vice versa. The serological nonresponders and treatment failure cases diagnosed at 12 months were given repeat treatment with the alternative drug.

## Evaluating latent syphilis treatment response

Treatment response was assessed with the decline in the RPR titers at the 12$^{th}$ and 24$^{th}$ months posttreatment. The repeated treatment cases had RPR titer determinations 12 months later. An adequate serologic response was defined with seroreversion (reactive to nonreactive RPR) or > 4-fold decline in the serum RPR titer (serofast). Serological nonresponse was described with the RPR titers decreasing < 4-fold and treatment failure with > 4-fold increase in the RPR titers. While evaluating the treatment response, we interviewed for possible behavioral risk factors that would make the study participant prone to reinfection. In the absence of such data, the treatment response was interpreted as serological treatment failure.

## Statistical analysis

We employed descriptive statistical methods to analyze the demographics and other characteristics of the participants. Qualitative data were presented as numbers with proportions. Fisher's exact test assessed associations between categorical variables. Quantitative data with Gaussian distributions were presented as means with standard deviations or medians with interquartile ranges (IQR) when the data showed a skewed distribution. Bivariate and multivariable logistic regression analyses were used to assess the association between the different risk factors and the diagnosis of latent syphilis. The IBM SPSS Statistics software package, version 23.0 (IBM Corporation, Armonk, NY, USA), was used for statistical analysis. A two-tailed t-test was performed to assess statistical significance between groups, and p-values < 0.05 were considered statistically significant.

## Results

We enrolled 823 PWH, receiving cART without clinically manifested syphilis (Fig 1). The mean age was 41.7 (SD = 10) years, 533 (64.8%) were females, and 290 (35.2%) were males (sex ratio about 2:1). Of the study participants, 44% were married, 25% had a history of sexually transmitted infection (STI), and 52% practiced condom use. Ninety percent were taking a first-line cART regimen. Fifty of the study participants (6%) had failed the first-line cART regimen and thus had HIV treatment failure (Table 1). The study group's median duration of cART was for 72 months (IQR: 36–108), and the median CD4 count was 431 cells/mm³ (IQR:292–600) (Table 1).

### Biologic—False Positive RPR (BFP-RPR) prevalence

With the initial qualitative RPR test, 46 (5.6%) tested reactive. Eight of the 46 reactive RPR sera had a weak reaction observed, and 15 out of the 46 sera tested nonreactive with TPHA (Fig 1). Thus, the prevalence of BFP-RPR varied from 18.4% (7/38, excluding the weakly reactive RPR sera) to 32.6% (15/46, including the weakly reactive RPR sera).

### Risk factors for latent syphilis

Of the 31 cases diagnosed with latent syphilis (reactive sera for RPR and TPHA), 22 were men (71%), and nine were women (29%) (Table 2). Thus, the prevalence of latent syphilis infection in the study population was 3.8% (95% CI, 2.6–5.3). The latent syphilis group, compared to the no latent syphilis group, was older by 6.5 years (mean age 48 years versus 41.5 years), had more history of STI, and had a longer duration on cART (Table 2). Although statistically not significant (p = 0.1), the proportion of cases with HIV treatment failure was two times higher (12.9% vs. 5.8%) in the latent syphilis group (Table 2). In bivariate analysis, risk factors associated with latent syphilis risk were male sex, increasing age, history of STI, and months on cART (Table 2). Multivariable logistic regression analysis revealed the odds of latent syphilis infection rose with the male sex (aOR = 3.14,95% CI 1.35–7.33), increasing age (aOR = 1.04/

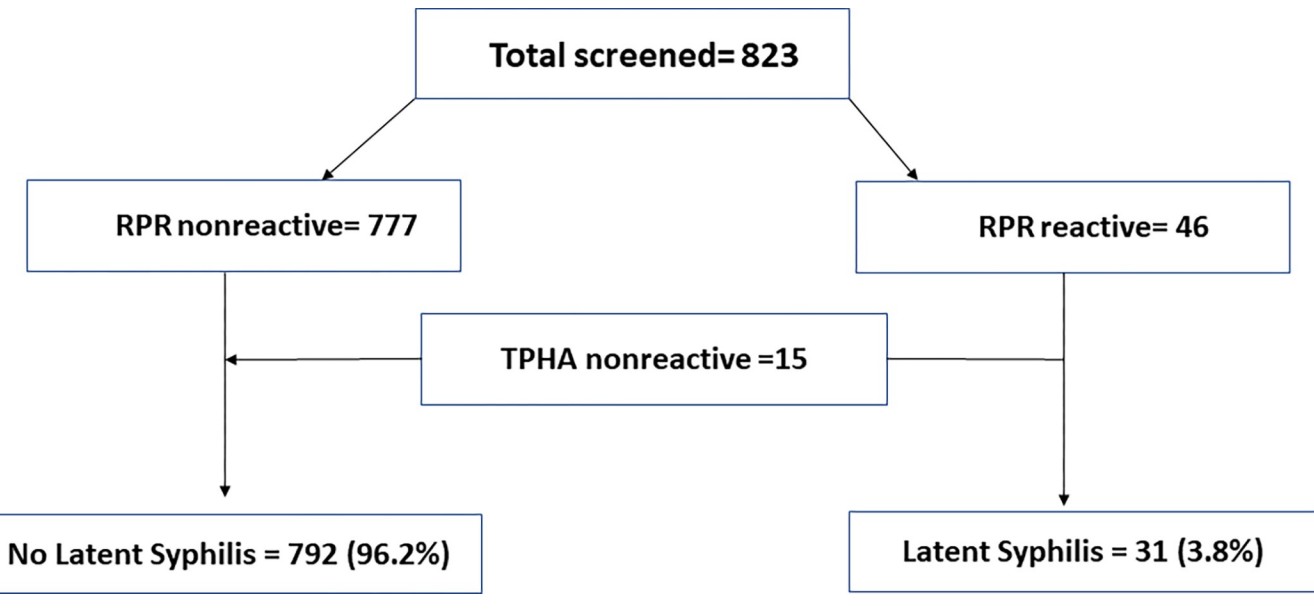

**Fig 1.**

**Table 1. Demographic and clinical characteristics of the 823 participants at baseline.**

| Characteristic | Result |
|---|---|
| Female sex–no.(%) | 533 (64.8%) |
| Age in years- mean (SD) | 41.7(10) |
| Marital status- no. (%) | |
| Single | 121(14.7%) |
| Married | 365(44.3%) |
| Divorced | 166(20.2%) |
| Widow/er | 171(20.8%) |
| Educational status- no. (%) | |
| Illiterate | 114(13.9%) |
| Primary | 332(40.3%) |
| Secondary | 262(31.8%) |
| Higher education | 115(14%) |
| History of any STI, self-no. (%) | 203 (24.7%) |
| History of any STI in the partner- no (%) | 62(7.5%) |
| New sex partner in the last three months- no. (%) | 26 (3.2%) |
| Any practice of condom use- no. (%) | 392(47.6%) |
| Any history of blood transfusion- no. (%) | 92(11.2%) |
| Sexual partner confirmed ever HIV positive- no. (%) | 331(40.2%) |
| Pre cART CD4 count- median (IQR) | 193(117–310) |
| The last CD4 count- median (IQR) | 431(292–600) |
| Months on cART-median (IQR) | 72(36–108) |
| HIV-RNA level not suppressed- no. (%) | 50 (6%) |
| Currently on First-line cART regimen- no. (%) | 740(89.9%) |
| Zidovudine/Lamivudine +Efavirenz | 7 (0.95%) |
| Tenofovir/Lamivudine/Efavirenz | 134 (18.1%) |
| Tenofovir/Lamivudine/Dolutegravir | 593 (80.1%) |
| Abacavir+Lamivudine+Efavirenz | 6 (0.8%) |
| Currently on Second-line cART regimen- no. (%) | 83(10.1%) |
| Zidovudine/Lamivudine+Atzanavir/ritonavir | 39 (47%) |
| Tenofovir/Lamivudine+Atanazavir/ritonavir | 36 (43%) |
| Abacavir/Lamivudine + Atazanavir/ritonavir | 8 (10%) |
| Headache +/- any neurological symptoms- no. (%) | 49(6%) |

no. = number, SD = standard deviation, IQR = interquartile range, cART = combination antiretroviral therapy, STI = sexually transmitted infection

year, 95% CI 1.00–1.08), and the number of months on cART (aOR = 1.01/month, 95% CI 1.00–1.01) (Table 3).

## Clinical and laboratory characteristics of the latent syphilis cases

We systematically investigated the 31 subjects diagnosed with latent syphilis by reviewing their medical records and assessing symptoms. The cardiovascular and nervous systems were carefully examined. Only three complained of headaches. The systematic physical examination was not revealing. The baseline RPR titers values were as follows: ≤1:4 in 19 (61.3%), between 1:8 and 1:32 in 10 (32.2%), and above 1:32 in 2 (6.4%). The latter two sera had RPR titer values of 1:256 and 1:1024 and HIV RNA levels of 4 logs/mL and 6 logs/mL, respectively (Table 4).

**Table 2. Demographics and clinical characteristics of the latent syphilis cases versus not.**

| Variables | Latent syphilis Total = 31 | No Latent syphilis Total = 792 | p-value |
|---|---|---|---|
| Male sex- no. (%) | 22 (71%) | 279(33.8%) | <0.001 |
| Age in years- mean (SD) | 48(12.7) | 41.5(9.8) | <0.001 |
| Multiple partners- no. (%) | 1(3%) | 61(7.7%) | 0.35 |
| New partner-no. (%) | 2(6.5%) | 24(3%) | 0.29 |
| History of any STI, self-no. (%) | 15(48.4%) | 188(23.7%) | 0.002 |
| History of any STI in the partner- no. (%) | 5(16.1%) | 57(7.2%) | 0.02 |
| Any history of blood transfusion- no. (%) | 3(9.7%) | 89(11.2%) | 0.78 |
| Any practice of condom use- no. (%) | 11(35.5%) | 381(48.1%) | 0.17 |
| Months on cART-median (IQR) | 96(60–144) | 72(36–108) | 0.02 |
| Pre cART CD4 count-median (IQR) | 178(112–256) | 193(117–312) | 0.54 |
| Last CD4 count- median (IQR) | 379(249–530) | 432(295–606) | 0.19 |
| HIV treatment failure- no. (%) | 4(12.9%) | 46(5.8%) | 0.1 |
| Second-line cART regimen- no. (%) | 5(16.1%) | 78 (9.8%) | 0.26 |

## Serologic response to treatment

Two cases randomly assigned to the BPG group refused the injection, so they were treated with DOXY. None of the DOXY assigned declined to take the treatment. Finally, 13 (41.9%) received BPG, and 18 (58.1%) received DOXY. We did not observe any severe adverse drug reactions. In the 12[th] month, the treatment outcomes were: 27 (87.1%) had an adequate serologic response (17 DOXY,10 BPG), 3 (9.7%) were serological nonresponders (2 BPG, DOXY), and one (3.2%) treatment failure (BPG) (Table 4). The adequate serologic response at 12 months comparing BPG with DOXY showed a non-significant difference (p = 0.37, Fisher's exact test). The three serological nonresponders and one treatment failure case treated for the second time with the alternative drug had posttreatment RPR titers determined after 12 months. The RPR titers of these cases were interpreted as follows: one had an adequate serologic response (RPR titer 1:16), two were serofast (RPR titer 1:4), and one had an RPR titer value of 1:2 repeatedly; likely to be BFP-RPR.

## Investigation for neurosyphilis

Lumbar punctures (LP) were performed in seven cases diagnosed with latent syphilis. The indications were syphilis treatment failure (3/7), RPR titer $\geq$ 1:32 (2/7) and peripheral CD4 cell count $\leq$ 350 cells/mm$^3$ (2/7 cases). Only one of the seven cases complained of long-standing headaches. None of the cases examined fulfilled the diagnostic criteria of neurosyphilis (positive CSF VDRL or the presence of CSF WBC > 10 cells/mm$^3$).

**Table 3. Multivariable logistic regression modeling for latent syphilis.**

| Variable | OR | 95% CI | p-value | aOR | 95%CI | p-value |
|---|---|---|---|---|---|---|
| Male sex | 4.78 | 2.17–10.52 | <0.001 | 3.14 | 1.35–7.33 | 0.008 |
| Age in years | 1.06 | 1.03–1.10 | <0.001 | 1.04 | 1.00–1.08 | 0.041 |
| History of any STI | 3.01 | 1.46–6.29 | 0.003 | 2.01 | 0.94–4.32 | 0.072 |
| Months on cART | 1.01 | 1.00–1.01 | 0.028 | 1.01 | 1.00–1.01 | 0.045 |
| HIV treatment failure | 2.4 | 0.80–7.16 | 0.12 | 1.84 | 0.57–5.92 | 0.304 |

OR = odds ratio, aOR = adjusted odds ratio

**Table 4. Demographic, clinical, and laboratory characteristics of the 31 latent syphilis cases.**

| No | Age | Sex | Persistent neurological symptoms or signs | Log HIV-RNA level | CD4 count cells/mm³ | Baseline RPR titer | LP | Initial Treatment | RPR titer after 12 months | RPR titer after 24 months |
|----|-----|-----|-------------------------------------------|-------------------|---------------------|--------------------|----|-------------------|---------------------------|---------------------------|
| 1 | 64 | M | No | 0 | 530 | 1:16 | No | DOXY | NR | NR |
| 2 | 57 | M | No | 0 | 327 | 1:16 | No | DOXY | 1:8 | 1:4 |
| 3 | 45 | F | No | 0 | 137 | 1:16 | No | DOXY | NR | NR |
| 4 | 30 | F | No | 0 | 356 | 1:16 | No | BPG | NR | NR |
| 5 | 49 | M | No | 0 | 511 | 1:4 | No | DOXY | NR | NR |
| 6 | 58 | M | No | 0 | 337 | 1:2 | No | BPG | NR | NR |
| 7 | 50 | M | No | 0 | 468 | 1:2 | No | BPG | NR | NR |
| 8 | 39 | M | No | 0 | 444 | 1:2 | No | DOXY | NR | NR |
| 9 | 47 | M | No | 0 | 637 | 1:4 | No | DOXY | NR | NR |
| 10 | 57 | M | No | 0 | 790 | 1:2 | No | DOXY | NR | NR |
| 11 | 48 | M | No | 0 | 569 | 1:4 | No | DOXY | NR | NR |
| 12 | 47 | F | No | 0 | 218 | 1:4 | No | DOXY | NR | NR |
| 13 | 60 | F | No | 0 | 692 | 1:4 | Yes | BPG | 1:16 | 1:4 |
| 14 | 60 | F | No | 0 | 693 | 1:2 | No | BPG | NR | NR |
| 15 | 48 | M | No | 0 | 256 | 1:8 | No | DOXY | NR | NR |
| 16 | 49 | M | No | 0 | 512 | 1:2 | No | BPG | NR | NR |
| 17 | 60 | M | No | 0 | 485 | 1:2 | No | DOXY | NR | NR |
| 18 | 40 | M | No | 6 | 131 | 1:256 | Yes | BPG | 1:64 | 1:16 |
| 19 | 32 | F | No | 0 | 570 | 1:4 | No | DOXY | NR | NR |
| 20 | 39 | M | No | 0 | 359 | 1:2 | No | BPG | NR | NR |
| 21 | 72 | F | No | 0 | 398 | 1:4 | No | BPG | NR | NR |
| 22 | 40 | M | No | 0 | 796 | 1:2 | No | DOXY | NR | NR |
| 23 | 68 | M | No | 0 | 303 | 1:2 | Yes | BPG | 1:2 | 1:2 |
| 24 | 51 | M | No | 0 | 293 | 1:16 | Yes | BPG | 1:8 | NR |
| 25 | 60 | M | Yes | 4 | 108 | 1:1024 | Yes | DOXY | 1:256 | NR |
| 26 | 22 | F | No | 0 | 347 | 1:16 | No | BPG | NR | NR |
| 27 | 27 | M | No | 0 | 287 | 1:16 | No | BPG | NR | NR |
| 28 | 37 | M | No | 4 | 153 | 1:2 | Yes | DOXY | NR | NR |
| 29 | 24 | M | No | 4 | 106 | 1:4 | Yes | DOXY | NR | NR |
| 30 | 56 | M | No | 0 | 623 | 1:8 | No | DOXY | NR | NR |
| 31 | 53 | F | No | 0 | 310 | 1:8 | No | DOXY | NR | NR |

NR = nonreactive, DOXY = doxycycline, BPG = benzathine penicillin G

## Discussion

Our findings suggest that the prevalence of latent syphilis in PWH receiving cART is 3.8% compared with 9.8% in 2013 by Eticha BT *et al.* [37] and 7.3% in 2015 by Shimeles T *et al.* [38]. All were hospital-based seroprevalence studies with different methods. The results, however, show a declining trend of syphilis seroprevalence in Ethiopian PWH. The national antenatal care-based syphilis surveillance report from 2007 to 2014 showed a similar declining trend [39]. On the contrary, there is no national data on PWH to compare the prevalence trends of syphilis [5]. The findings are consistent with an estimated syphilis seroprevalence in low-income countries, ranging from 3.5 to 4.6% [2].

In our study, the prevalence of BFP-RPR test results varied from 18.4% (without weakly reactive RPR) to 32.6% (with weakly reactive RPR). An Ethiopian study from 1999 to 2001 in 409 factory workers (10.8% HIV-infected) showed a BFP-RPR test result of 8.2% [40]. Thus,

the current BFR-RPR results are considerably higher than in previous reports. While this may be due to the quality of the serological tests used, the qualitative nature of the tests, and observer variations, other factors may also contribute to this result. First, although cART may reduce the odds of BFP-RPR, still HIV infection is a significant risk factor for syphilis [41]. Indeed in PWH, the prevalence of BFP-RPR is higher (4–15%) than in the general population (5%) [41, 42], noting an exaggerated prevalence of BFP-RPR (6–33%) occurs in HIV-infected intravenous drug users [43]. Second, low RPR titers (≤1:4) may persist after successful treatment of syphilis, contributing to a high rate of BFP-RPR results [44]. Third, increasing age and coinfections like hepatitis B and C might explain some BFR-RPR results [45].

The sensitivity of the RPR test varies from 98% in early latent syphilis to 73% in late latent syphilis [46]. False-negative RPR test results are bound to happen, and one reason is the pro-zone effect. It is described at any stage of syphilis, mainly when undiluted sera are analyzed [45]. Consequently, our study might have missed some latent syphilis cases as the qualitative RPR tests were performed with undiluted sera.

The baseline RPR titers among the latent syphilis cases were 1:4 or less in 19 (61%) samples, while only two (6.4%) had an RPR titer > 1:32. The other ten samples (32.2%) had RPR titer values between 1:4 and 1:32. The RPR titer values (<1:4) can be partly explained by BFP-RPR [45], whereas early syphilis may explain the two samples with RPR titer values above 1:32 [46]. The two subjects with RPR titers above 1:32 had HIV RNA levels of 4 logs/mL and 6 logs/mL. It supports early syphilis infection with an increased risk of HIV transmission [47]. Strikingly, syphilitic disease activity was low in the majority of our cases.

None of the seven cases with CSF analysis in our study had neurosyphilis. The published works quantifying neurosyphilis in HIV coinfected subjects in Africa are limited [33]. A Mozambique study evaluated 21 treatment-naive PWH and confirmed syphilis. All had CSF analysis at baseline, and four were diagnosed with asymptomatic neurosyphilis. However, none had a reactive CSF-RPR [48]. No cases of neurosyphilis were diagnosed in 31 HIV-syphilis coinfected Nigerians (29 early and two latent syphilis cases) [49]. One neurosyphilis case was identified in a South African study of 506 PWH [50]. This study did not mention the prevalence of syphilis and the number of subjects who underwent LP. A study from Tanzania of 167 treatment-experienced PWH with a 9.6% prevalence of syphilis managed to diagnose only one neurosyphilis patient [51]. The above studies had several limitations: small sample size, different diagnostic criteria for neurosyphilis, inconsistent exposure to cART, missing peripheral CD4 counts, and HIV RNA levels. Despite the limitations, all show a minimal risk of neurosyphilis with HIV coinfection. One explanation is the lower RPR titer values in the latent syphilis cases. The RPR titer values correlate with the risk of neurosyphilis in HIV coinfection [32]. An alternative explanation is that cART may mitigate neurological complications of syphilis [20]. In asymptomatic late syphilis cases, CSF abnormalities occur. However, the diagnosis of neurosyphilis was significantly associated with early syphilis in a multivariable model [52].

Men were significantly more affected with latent syphilis in the current study. Few other studies showed similar findings in Ethiopia. The African Cohort Study investigated the prevalence and risk factors associated with HIV and syphilis coinfection [53]. Four sub-Saharan African countries participated in this study between January 2013 and March 2020. The serologically confirmed syphilis cases (early and late syphilis) had a prevalence of 3%, the mean age was 38 years, and 58.6% were women [53]. Studying the sexual attitudes of our study participants, the health-seeking behavioral differences between the two groups and the effect of routine screening for syphilis during antenatal care partly explain the differences between the current study and previous reports.

Advancing age was another risk factor for latent syphilis infection. The odds of having latent syphilis increase by 4% for every year of life added. It is consistent with the study done by Shimeles T *et al*. [38]. Although it was not statistically significant, Eticha BT *et al*. [37] demonstrated a trend toward male sex and older age. Our study showed that PWH who were older and had a longer duration of cART had increased syphilis prevalence. This finding is consistent with Hu *et al*. [23] and Stolte *et al*. [54] but contrary to Huang *et al*. from a Taiwanese study [55].

Our study revealed that 81% achieved seroreversion after 12 months of treatment. One study indicated that nearly 70% of the latent syphilis cases with HIV coinfection achieved seroreversion [56]. Evaluating treatment effectiveness with the adequacy of serological responses showed a non-significant difference between BPG and DOXY, suggesting that DOXY can be used alternatively. All were adequate serologic responders at the 24th month, except one had a persistently positive RPR titer of 1:2, likely BFP-RPR.

The study limitations are: not being able to rule out reinfections, the reduced sensitivity of nontreponemal tests in late latent syphilis, missing asymptomatic neurosyphilis cases for we did not perform CSF analysis in all the latent syphilis cases, and the potential for false-negative test results due to prozone reactions may lead to underestimating the latent syphilis prevalence. The prevalent latent syphilis cases indicate previously untreated or inadequately treated syphilis infections. The antenatal history regarding screening and treatment of syphilis in women who gave birth was unavailable. Thus, our study does not show the incidence of syphilis and the risk of progression to latent infection with cART.

In conclusion, the study confirmed the relevance of screening for latent syphilis in PWH receiving cART. The current guidelines for PWH recommend syphilis screening for symptomatic and newly diagnosed, leading to high numbers of undiagnosed infections. Thus, we recommend annual screening for asymptomatic sexually active PWH. The risk of latent syphilis was higher in men than in women. Therefore, we recommend further investigations to address the sexual practices and the impact of routine antenatal screening. Notably, syphilis disease activity reduces during the latent phase of the disease. Accordingly, the risk of neurosyphilis in asymptomatic PWH with latent syphilis receiving cART is substantially low, suggesting a minor role for routine CSF analysis. DOXY is an alternative to BPG, and cART improves syphilis treatment response.

## Supporting information

**S1 Data.**
(XLSX)

## Acknowledgments

We thank the study participants and the HIV outpatient clinics staff of TASH, ARH, THC, and KHC. In addition, our thanks extend to the MEPI office at the College of Health Sciences, Addis Ababa University, for supporting the study.

## Author Contributions

**Conceptualization:** Selamawit Girma, Wondwossen Amogne.

**Data curation:** Selamawit Girma, Wondwossen Amogne.

**Formal analysis:** Selamawit Girma, Wondwossen Amogne.

**Funding acquisition:** Selamawit Girma.

**Investigation:** Selamawit Girma, Wondwossen Amogne.

**Methodology:** Selamawit Girma, Wondwossen Amogne.

**Supervision:** Wondwossen Amogne.

**Writing – original draft:** Selamawit Girma, Wondwossen Amogne.

**Writing – review & editing:** Selamawit Girma, Wondwossen Amogne.

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
