## [Decision Letter · Decision Letter 0]

24 Nov 2021

PONE-D-21-26734Latent syphilis in HIV treatment-experienced Ethiopians: Investigating risk factors, neurosyphilis, and response to therapy.PLOS ONE

Dear Dr. Amogne,

Thank you for submitting your manuscript to PLOS ONE. After careful consideration, we feel that it has merit but does not fully meet PLOS ONE’s publication criteria as it currently stands. Therefore, we invite you to submit a revised version of the manuscript that addresses the points raised during the review process.

We look forward to receiving your revised manuscript.

Kind regards,

Giordano Madeddu

Academic Editor

PLOS ONE

2. Thank you for stating the following in the Financial Section of your manuscript:

“The research was supported by a Medical Education Partnership Initiative (MEPI) grant for junior faculty members (D43TW010143) obtained from the US National Institutes of Health, Fogarty International Center. The funder had no role in the study design, conduct, data collection, analysis, publication decision, or manuscript preparation. The grant received was to conduct the study and does not cover expenses related to the manuscript preparation and publication. “

Reviewers' comments:

Reviewer's Responses to Questions

**Comments to the Author**

1. Is the manuscript technically sound, and do the data support the conclusions?

Reviewer #1: Yes

Reviewer #2: Yes

2. Has the statistical analysis been performed appropriately and rigorously? 

Reviewer #1: Yes

Reviewer #2: Yes

3. Have the authors made all data underlying the findings in their manuscript fully available?

Reviewer #1: Yes

Reviewer #2: No

4. Is the manuscript presented in an intelligible fashion and written in standard English?

Reviewer #1: Yes

Reviewer #2: No

5. Review Comments to the Author

Reviewer #1: The study presents results of original study. The study objectives were clearly set out. Methodology, statistics were performed with high technical standards and adequately described. The conclusions are appropriately supported with data. The paper is well written in standard English.

In the background, addition of syphilis prevalence in the area of study or Ethiopian general population or pregnant women would give context to the reader.

Reviewer #2: Full title: Latent syphilis in HIV treatment-experienced Ethiopians: investigating risk factors, neurosyphilis, and response to therapy. 4 Short title: Latent syphilis in HIV coinfected Ethiopians

Many typos are present in the text, such as capital letters (e.g., line 61 “Syphilis”, and grammar mistakes. Please re-read the manuscript carefully and fix them.

Abbreviations should be written entirely in the first appearance in the text (e.g., SD)

Title

I suggest removing “neurosyphilis” from the title.

I suggest modifying the short title in Latent syphilis in Ethiopians living with HIV

Introduction

I suggest deleting “Syphilis is a major global health problem.” and starting with “The World Health […].

Many sentences seem to be disconnected from the rest of the text (e.g. lines 44 “Neurosyphilis risk increases with HIV coinfection”). I suggest you read the manuscript carefully and try to create a more flowing text.

Furthermore, I suggest you add that people with HIV that have a viral load <200copies/mL and a STD still does not transmit HIV, adding these two references https://doi.org/10.1097/QAD.0000000000002825 and https://doi.org/10.1016/S0140-6736(19)30418-0.

The authors wrote, “Frequently PWH are diagnosed with latent syphilis without being aware of prior symptomatic disease” I suggest not using “frequently,” but write the percentage if you have one.

Methods

The authors wrote, “The indications were RPR titer > 1:32, peripheral blood CD4 count < 350 cells/mm3 if the HIV RNA level is above 200 copies/mL” and “We included the HIV RNA level criterion to exclude immunological nonresponders (a category of HIV-infected patients on cART who have suppressed HIV viremia but a suboptimal increase in their CD4 cell count)”.

This part is unclear. Neurosyphilis could also be present in people with more than 350 CD4/mm3. Please comment.

The definition of neurosyphilis, in my opinion, is not correct. The authors wrote “. Neurosyphilis was diagnosed with a reactive cerebrospinal fluid (CSF) venereal disease research laboratory (VDRL) test result or the presence of CSF pleocytosis (WBC > 10 cells/mm3)”. Therefore, I suppose that a patient with negative CSF VDRL but 11 WBC in the CFS was considered positive to neurosyphilis, and it could represent an important bias that could cause an overrate of neurosyphilis. Please comment.

Have patients with a diagnosis of latent syphilis performed an ophthalmic evaluation? Please comment

The authors wrote “The latent syphilis cases were assigned to receive treatment with three weekly intramuscular injections of BPG 2.4 million units”. I think that they mean one intramuscular injection for three weeks. Please comment.

Tables. Please add a caption in each table explaining the abbreviations’ meaning.

Table 1. Please write the number of people included in the study in the title. Instead of using the ratio of male/female, write the percentage of them. Please modify “History of STI, self (n,%)” with “previous STI diagnosis (n,%).

It is not clear what “Practice condom use, any (n,%)” means. Please comment

The author wrote “Baseline CD4 count (median, IQR)”. However, it is not clear what baseline means.

6. PLOS authors have the option to publish the peer review history of their article (what does this mean?). If published, this will include your full peer review and any attached files.

Reviewer #1: No

Reviewer #2: No

---

## [Decision Letter · Decision Letter 1]

24 Mar 2022

PONE-D-21-26734R1Latent syphilis in HIV treatment-experienced Ethiopians and response to therapy.PLOS ONE

Dear Dr. Amogne,

Thank you for submitting your manuscript to PLOS ONE. After careful consideration, we feel that it has merit but does not fully meet PLOS ONE’s publication criteria as it currently stands. Therefore, we invite you to submit a revised version of the manuscript that addresses the points raised during the review process.

We look forward to receiving your revised manuscript.

Kind regards,

Giordano Madeddu

Academic Editor

PLOS ONE

Reviewers' comments:

Reviewer's Responses to Questions

**Comments to the Author**

1. If the authors have adequately addressed your comments raised in a previous round of review and you feel that this manuscript is now acceptable for publication, you may indicate that here to bypass the “Comments to the Author” section, enter your conflict of interest statement in the “Confidential to Editor” section, and submit your "Accept" recommendation.

Reviewer #2: (No Response)

2. Is the manuscript technically sound, and do the data support the conclusions?

Reviewer #2: Partly

3. Has the statistical analysis been performed appropriately and rigorously? 

Reviewer #2: N/A

4. Have the authors made all data underlying the findings in their manuscript fully available?

Reviewer #2: No

5. Is the manuscript presented in an intelligible fashion and written in standard English?

Reviewer #2: No

6. Review Comments to the Author

Reviewer #2: The authors have not uploaded the answers' to my previous revision. For this reason, I am not able to perform this review.

7. PLOS authors have the option to publish the peer review history of their article (what does this mean?). If published, this will include your full peer review and any attached files.

Reviewer #2: No

---

## [Author Response · Author response to Decision Letter 1]

16 May 2022

Greetings,

We want to thank you for the valuable comments the reviewers gave us to improve the quality of our manuscript. As a result, we did several editorial works to ensure the clarity and continuity of our work. The revised manuscript is attached for your comments. We have also made the raw data available. Below you will find our response to the questions raised by the second reviewer. 

Response to the questions raised by reviewer 2:

Question #1: The authors wrote, "The indications were RPR titer > 1:32, peripheral blood CD4 count < 350 cells/mm3 if the HIV RNA level is above 200 copies/mL," and "We included the HIV RNA level criterion to exclude immunological nonresponders (a category of HIV-infected patients on cART who have suppressed HIV viremia but a suboptimal increase in their CD4 cell count)". This part is unclear. Neurosyphilis could also be present in people with more than 350 CD4+ cells/mm3. Please comment.

Authors' response: We accept the section on "diagnosis of neurosyphilis" was not explicit. Therefore, it is written again (refer to page 7). The indications for CSF analysis in our study were a/ serum RPR titer > 1:32 b/peripheral blood CD4 count < 350 cells/mm3, and HIV RNA level above 2.3 log/mL (or 200 copies/mL) c/serological nonresponse or treatment failure after latent syphilis therapy and d/unexplained persistent neurologic, ocular or otic symptoms or signs. CNS invasion with Treponema pallidum in PWH can occur at any stage of syphilis, CD4 count, and RPR titration. It may be asymptomatic or manifests with central nervous system affection symptoms. Asymptomatic neurosyphilis, however, rarely occurs in antiretroviral naïve PWH and syphilis confection with RPR titer values below 1:16 and > 350 peripheral CD4+ cells/mm3. Therefore, it is controversial that this group should require routine CSF evaluation. Several studies have suggested that serum RPR titers > 1:32 or CD4 counts < 350 in treatment naïve HIV-syphilis coinfected patients indicate a higher risk of asymptomatic infection. Thus, the current guidelines recommend evaluating the CSF for neurosyphilis based on these suggestions. The guidelines fail to address the risk of asymptomatic neurosyphilis in PWH with suppressed HIV RNA levels due to combination antiretroviral therapy (cART). In patients with CD4 counts > 350 and RPR titer > 1:32, CSF evaluation is highly recommended regardless of symptoms. There is a potential to miss asymptomatic cases presenting without symptoms and an RPR titer < 1:32 unless CSF analysis is done routinely. Since we didn't perform CSF analysis in all our latent syphilis cases, there is a potential missing, even though it is rare. It is one of our study limitations (page 13, lines 270-71). We didn't expect the RPR titer to show a favorable response with either BPG or DOXY therapy when there is neurosyphilis. It was the reason for including serological nonresponders and treatment failure cases for CSF analysis to rule out neurosyphilis. We conclude that it is quite rare for neurosyphilis to occur without symptoms in PWH on suppressive cART, CD4 above 350, and latent syphilis coinfection, provided the RPR titer value is below 1:32. 

Question #2: The authors wrote, "Neurosyphilis was diagnosed with a reactive CSF VDRL test or the presence of pleocytosis (WBC > 10 cells/mm3)". Therefore, I suppose that a patient with negative CSF VDRL but 11 WBC in the CSF was considered positive for neurosyphilis. It could represent an important bias that could cause an overrate of neurosyphilis. Please comment. 

Have patients with a diagnosis of latent syphilis performed an ophthalmic evaluation? Please comment.

Authors' response. The diagnosis of neurosyphilis is a complex process. CSF WBC count > 10 cells/mm3 is often used as a diagnostic criterion for neurosyphilis in PWH. However, the specificity is lower in PWH, for there are other pleocytosis causes outside of neurosyphilis. If this criterion alone is used, it will certainly lead to an overdiagnosis of neurosyphilis, particularly in PWH without symptoms. The level of uncertainty would decrease if both criteria are used together (CSF VDRL and WBC > 10 cells/mm3); however, the problem is the CSF VDRL lacks sensitivity. Even if using this criterion, none of the seven CSF we analyzed supported the diagnosis of neurosyphilis, implying syphilis disease activity is reduced in the latent stage. We conclude that neurosyphilis from latent syphilis infection is rare in PWH receiving virus suppressive cART and RPR titers below 1:32. 

We did ophthalmic examination in cases diagnosed with latent syphilis ( page 6, lines 121-122).

---

## [Decision Letter · Decision Letter 2]

20 Jun 2022

Latent syphilis in HIV treatment-experienced Ethiopians and response to therapy.

PONE-D-21-26734R2

Dear Dr. Amogne,

We’re pleased to inform you that your manuscript has been judged scientifically suitable for publication and will be formally accepted for publication once it meets all outstanding technical requirements.

Kind regards,

Giordano Madeddu

Academic Editor

PLOS ONE

Additional Editor Comments (optional):

Reviewers' comments:

Reviewer's Responses to Questions

**Comments to the Author**

1. If the authors have adequately addressed your comments raised in a previous round of review and you feel that this manuscript is now acceptable for publication, you may indicate that here to bypass the “Comments to the Author” section, enter your conflict of interest statement in the “Confidential to Editor” section, and submit your "Accept" recommendation.

Reviewer #2: All comments have been addressed

2. Is the manuscript technically sound, and do the data support the conclusions?

Reviewer #2: Yes

3. Has the statistical analysis been performed appropriately and rigorously? 

Reviewer #2: Yes

4. Have the authors made all data underlying the findings in their manuscript fully available?

Reviewer #2: Yes

5. Is the manuscript presented in an intelligible fashion and written in standard English?

Reviewer #2: Yes

6. Review Comments to the Author

Reviewer #2: the authors have adequately addressed my comments, and in my opinion the manuscript is now suitable to be publish

7. PLOS authors have the option to publish the peer review history of their article (what does this mean?). If published, this will include your full peer review and any attached files.

Reviewer #2: No

---

## [Editor Report · Acceptance letter]

4 Jul 2022

PONE-D-21-26734R2 

Latent syphilis in HIV treatment-experienced Ethiopians and response to therapy. 

Dear Dr. Degu:

I'm pleased to inform you that your manuscript has been deemed suitable for publication in PLOS ONE. Congratulations! Your manuscript is now with our production department. 

Kind regards, 

on behalf of

Dr. Giordano Madeddu 

Academic Editor

PLOS ONE